# Isolation of Cherry Seed Oil Using Conventional Techniques and Supercritical Fluid Extraction

**DOI:** 10.3390/foods12010011

**Published:** 2022-12-20

**Authors:** Ivana Dimić, Branimir Pavlić, Slađana Rakita, Aleksandra Cvetanović Kljakić, Zoran Zeković, Nemanja Teslić

**Affiliations:** 1Faculty of Technology, University of Novi Sad, Blvd. cara Lazara 1, 21000 Novi Sad, Serbia; 2Institute of Food Technology, University of Novi Sad, Blvd. cara Lazara 1, 21000 Novi Sad, Serbia

**Keywords:** sour cherry seed oil, *Prunus cerasus* L., extraction techniques, chemical composition, antioxidant activity

## Abstract

This study aims to compare the suitability of three extraction techniques (cold pressing, Soxhlet and supercritical fluid extraction (SFE)) to isolate oil from cherry seeds. Oils were examined in terms of extraction yield, fatty acids profile, tocopherols yield and antioxidant activity. Additionally, influence of SFE parameters was evaluated using one-factor-at-a-time design with pressure (200–350 bar), temperature (40–70 °C), flow rate (0.2–0.4 kg/h) and particle size (<800 µm and >800 µm). Oil yields ranged from 2.50% to 13.02%, whereas the highest yield was achieved with SFE. Samples were rich in polyunsaturated fatty acids, regardless of the applied extraction technique. The main fatty acids were linoleic (46.32–47.29%), oleic (40.89–41.65%), palmitic (6.56–8.00%) and stearic (2.21–2.30%) acid. Total tocopherols yield was between 16.63 mg/100 g oil and 60.61 mg/100 g oil, and highest yield was achieved with SFE. Among the tocopherols, γ-tocopherol was the most abundant, followed by α-, δ- and β-tocopherol. Antioxidant activity was determined using 1,1-diphenyl-2-picrylhydrazyl (DPPH) and 2,2’-azinobis-(3-ethylbenzothiazoline-6-sulfonic) cation (ABTS) assays, and the results indicated that SFE extracts exhibited better or comparable antioxidant potential compared to traditional techniques. The comparison between modern and conventional extractions for oil recovery demonstrates pros and cons for the possibility of industrial application.

## 1. Introduction

The expansion of the world population has caused higher demand for food, resulting in the constantly expanding production of various fruits, vegetables and cereals. Consequently, with increased food processing, there has been a significant growth in agricultural waste leading to adverse environmental effects and economic losses. According to the latest research, food by-products are an affordable natural resource rich in bioactive compounds having a positive health impact, though they are not adequately exploited. Nowadays, people’s awareness of the benefits of a healthy diet has increased, and this awareness has brought up the question of including high-added-value food products to their daily routine [1,2,3].

Sour cherry (*Prunus cerasus* L.) represents a part of the Rosaceae family, and the majority of the species originates from Europe and Asia. This food has been recognized as ’super-food’ for its valuable impact on human health, since it may be helpful in the treatment of cancer and neurodegenerative diseases [4,5]. This fruit is rarely consumed fresh; on the contrary, it is processed into juices, jams, marmalades, toppings, alcoholic drinks and processed fruit products [6]. The approximate amount of sour cherry production reached 1,479,045 tons in the world in 2020 [7], which leads to the generation of a significant amount of by-products.

The process of recovering oils and lipids from various nutritive and non-nutritive resources [8] has been introduced in various industries, such as cosmetics, elastomer and chemicals. Seeds have been identified as a great source of oil, rich in bioactive compounds having antioxidant, antimicrobial and anti-inflammatory effects [9]. The oil content of sour cherry kernels, when the outer shell is eliminated, is estimated to be 17.0%, similar to soybeans and olives. It is rich in polyunsaturated fatty acids, especially linoleic and linolenic acid. Among monounsaturated fatty acids, oleic acid is present to the greatest extent. Sour cherry seed oil is known to be a resource of vitamins and provitamins, such as tocopherols and carotenoids [2]. Additionally, the oil contains phenolic compounds, sterols, triglycerides and diglycerides. Among phenolic compounds, anthocyanins, flavan-3-ols and hydroxycinnamic acids comprise the majority [6,10].

Throughout history, different traditional techniques to recover oil and bioactive compounds from plants were employed, and these are known as cold-pressing, maceration, percolation, steam distillation, hydrodistillation, solvent extraction and Soxhlet extraction. Cold pressing is a mechanical technique performed in the absence of heat and chemical solvents, which results in obtaining extracts with higher antioxidant potential [11,12]. Other techniques usually include usage of organic solvents that are often detected in traces in the product or include higher temperatures. The latter could result in thermally unstable bioactive compounds being decomposed and overall extract yield reduced [13]. Such limitations have increased the interest of scientists to conduct research with a view of developing modern risk-free and eco-friendly techniques, often referred to as green extraction techniques [14]. The modern extraction techniques have been developed to satisfy industrial demands for lowering solvent consumption, cutting down extraction time and increasing extract yield, while simultaneously maintaining its high quality. In previous years, the most frequently studied extraction techniques from the perspective of industrial use were microwave-assisted extraction (MAE), ultrasound-assisted extraction (UAE), supercritical fluid extraction (SFE), pressurized liquid extraction (PLE), pulsed electric field assisted extraction (PEF), enzyme-assisted extraction (EAE) and deep eutectic solvents (DES) extraction. Among different techniques known for their simplicity, non-conventional solvent use, safety and improved extract quality, supercritical fluid extraction (SFE) has been found to have potential use in the food, nutraceutical and pharmaceutical industry [15,16].

The SFE process was primarily introduced to extract caffeine from coffee beans and α-acids from hops [12]. Supercritical CO_2_ is convenient for use since it can be obtained in high purity, and it is inexpensive, harmless, non-toxic and non-flammable. It does not leave residues in the extract since it can be removed by depressurization [11]. Carbon dioxide stands out for its advantageous critical conditions (pressure of 7.38 MPa and temperature of 31.1 °C), making it suitable for the extraction of thermally unstable or corrodible compounds since it is performed in the absence of oxygen. The major disadvantage of CO_2_ is low polarity, which is useful for the extraction of the lipophilic compounds, but it is not desirable for the extraction of polar substances. This drawback has been effectively resolved by the addition of modifiers, such as water, methanol, ethanol, acetone and acetonitrile. In this way, it is possible to adjust the polarity and selectivity of the mixture and aim the extraction towards the isolation of the desired compounds [12,17].

Fluid can be converted to a supercritical state when its pressure and temperature exceed the critical point and liquid and gas state become homogeneous. In such a state, fluid possesses the properties of both phases; it has gas-like diffusivity and liquid-like solvating power [9]. Slight modifications of pressure and temperature can alter the characteristics of the supercritical fluid, making it appropriate for selective bioactives recovery [18]. With pressure increase, supercritical fluid has higher density and can easily diffuse into plant material, which enhances the extraction rate. On the other hand, temperature increase creates a rather contradictory effect since the volume of the fluid expands, the density becomes lower and solvating power weakens. At the same time, solute vapor pressure is enhanced. Once these two phenomena become consistent, an effect named “crossover point” occurs. The existence of the crossover point shows the importance of experimental planning in gaining insight into the interaction of pressure and temperature and their impact on the total extraction yield [19]. Additionally, it is important to emphasize the flow rate influence since its increase enhances the concentration gradient, and it is possible to achieve higher extraction yield of target compounds. On the other hand, high flow rate results in short contact time between the plant material and the solvent which is insufficient to perform successful extraction [20].

Due to the lack of data related to sour cherry seed oil, the main goal of this study is to examine the influence of different SFE parameters (pressure, temperature, flow rate and particle size) on the extraction of various bioactives from this food by-product. Samples were compared based on the fatty acid profile, tocopherol content and antioxidant activity. Thus, further optimization of the SFE process and its commercial application could be implemented in order to obtain the extract with the highest yield of particular bioactive compounds.

## 2. Materials and Methods

### 2.1. Plant Material

The by-product of cherry seeds was obtained from the local cold-pressed oil factory, PAN-UNION d.o.o. (Novi Sad, Serbia). Cherry seeds were firstly milled in a hammer mill (ABC Engineering, Pančevo, Serbia) and sieved through the vibro-sieve set (CISA Cedaceria Industrial, Barcelona, Spain) to calculate mean particle size. The mean particle size of the milled cherry seeds used for the experimental work was 741 μm. The raw material was afterwards subjected to fractionation on the same vibro-sieve set, and the size of obtained fractions was <800 μm and >800 μm. The fractions were later used to investigate the influence of particle size on the SFE process.

### 2.2. Reagents

Helium (>99.9997%) and carbon dioxide (99.9%) were purchased from Messer Technogas A.D., Novi Sad. *n*-Hexane was purchased from Merck KgaA, Darmstadt, Germany. Ethanol was purchased from Sani-Hem D.O.O., Novi Bečej, Serbia. Methanol was purchased from Lach-ner Ltd., Neratovice, Chech Republic. 1,1-Diphenyl-2-picrylhydrazyl-hydrate (DPPH) and 2,2′-azino-bis(3-ethylbenzothiazoline-6-sulfonic acid) diamonnium salt (ABTS) were purchased from Sigma Aldrich, St. Louis, MO, United States. Supelco 37 component FAMEs mix, DL-α tocopherol (99.9%), rac-β-tocopherol (99%), γ-tocopherol (97.3%) and δ-tocopherol (95.2%) were purchased from Supelco Inc., Bellefonte, PA, USA.

### 2.3. Extraction Techniques

#### 2.3.1. Cold Pressing

Cold pressing (CP) of the cherry seeds was performed on a screw press (SZR “Mikron”, Temerin, Serbia). The mass of 6 kg of cherry seeds was cold pressed to recover oil. The internal temperature was 70 °C, and the rotation speed was 26 Hz. The cold pressing was conducted in one repetition. Extracted oil was collected and stored in a tank, allowing sedimentation of fine particles. After a few days of sedimentation, oil was sampled from the upper part of the tank, transferred into amber glass bottles of 250 mL, filled to the top (to exclude the presence of O_2_ as much as possible) and stored at 4 °C until the analysis.

#### 2.3.2. Soxhlet Extraction

For the Soxhlet extraction (SE), 30.0 g of milled cherry seeds was extracted by adequate non-polar solvent in the 1:4 ratio using Soxhlet equipment. The used solvents were *n*-hexane and methylene chloride (120 mL). The extraction lasted for 6 h and was followed by evaporation of the solvent on a rotary evaporator. Both extractions were performed in one repetition. Obtained extracts were further dried for 24 h at room temperature and stored in amber glass bottles of 250 mL at 4 °C until further analysis.

#### 2.3.3. Supercritical Fluid Extraction

The supercritical fluid extraction (SFE) was performed on a laboratory scale high-pressure extraction plant (HPEP, NOVA-Swiss, Effretikon, Switzerland) described by Pavlić et al. [21]. Cherry seeds (130.0 ± 0.01 g) were placed in the extraction vessel. The experiments lasted for 4 h at different combinations of pressure (200, 275 and 350 bar), temperature (40, 55 and 70 °C) and CO_2_ flow rate (0.2, 0.3 and 0.4 kg/h), according to the one-factor-at-a-time (OFAT) design, while average sample particle size was constant (741 µm). This approach was used to evaluate the impact of SFE parameters on extracts’ chemical composition and extraction yield. The experimental plan consisted of 9 experiments and was later expanded with 2 additional runs to examine the influence of different sample particle size (<800 and >800 µm) on bioactive compounds yield and antioxidant activity. Sample names, process conditions and extraction yields are given in Table 1.

The extracts were collected in amber glass bottles of 250 mL and kept at 4 °C until further analysis. All the experiments were conducted in one repetition. Total extraction yield for all the performed extraction techniques was gravimetrically determined and calculated according to the following equation:(1)$$Y\;\left[\%\right]=\frac{mass\;of\;extracted\;oil}{mass\;of\;cherry\;seeds}\times 100$$

The mass of cherry seeds was different in the applied methods, due to the apparatus capacity. However, we were able to apply the same equation in all the yield calculations and express the yield in percents.

### 2.4. Chemical Analysis

#### 2.4.1. Fatty Acid Profile of the Oils

Fatty acid methyl esters were prepared from extracted lipids using a method based on 14% boron-trifluoride methanol solution, which is the recommended method for this type of sample [22]. Nitrogen was used for drying and removing the solvent from fatty acid (FA) methyl esters. Obtained samples were analyzed at GC Agilent 7890A system (Agilent Technologies, Santa Clara, CA, USA) with FID, automated liquid injection module, equipped with capillary column with fused silica gel (SP-2560, 100 m × 0.25 mm, I.D., 0.20 µm, Supelco Analytical, Bellefonte, PA, USA). The initial temperature was 140 °C with a hold of 5 min. Heating up to 240 °C was in 2 °C increments up, and the hold on 240 °C was 5 min. The injector and detector temperatures were set at 250 °C. Helium was used as a carrier gas (flow rate = 1.26 mL/min). Identification of the fatty acids was achieved based on the comparison of the retention times with retention times of the standards from Supelco 37 component FAMEs mix and data from the internal data library, according to the earlier experiments and GC/MS analysis. The results were expressed as mass of fatty acid or fatty acid group (g) per 100 g of oil.

Evaluation of functional quality of cherry seed oil was conducted by calculating three indices from the FA profile. The ratio between hypocholesterolemic and hypercholesterolemic FAs (*h*/*H*) was calculated according to the following equation [23]:(2)$$\frac{h}{H}=\frac{C18:1+C18:2+C18:3}{C14:0+C16:0}$$

The atherogenocity index (*AI*) and thrombogenicity index (*TI*) were calculated according to the following equations [23]:(3)$$AI=\frac{C14:0+4\left(C16:0\right)}{\sum MUFA+\sum \omega -3+\sum \omega -6}$$
(4)$$TI=\frac{C14:0+C16:0+C18:0}{0.5(\sum MUFA)+3\sum \omega -3+0.5\sum \omega -6+\left(\frac{\sum \omega -3}{\sum \omega -6}\right)}$$
where *C*14:0 is myristic acid, *C*16:0 is palmitic acid, *C*18:0 is stearic acid, *C*18:1 is oleic acid, *C*18:2 is linoleic acid, *C*18:3 is α-linolenic acid, Σ*MUFA* is the sum of monounsaturated fatty acids, Σ*ω*−3 is the sum of the polyunsaturated *ω*−3 fatty acids, and Σ*ω*−6 is the sum of the polyunsaturated fatty *ω*−6 acids.

#### 2.4.2. Tocopherols Content

Tocopherols content was measured using high-pressure liquid chromatography according to the modified method [24]. Samples were diluted in *n*-hexane and filtered through RC 0.45 μm syringe filter (Agilent Technologies Inc., Böblingen, Germany). The HPLC system (Agilent liquid chromatography series 1260) consists of a quaternary pump, an autosampler and a fluorescence detector (Agilent Technologies Inc., Böblingen, Germany). Tocopherols separation was achieved using a normal-phase analytical column Luna^®^ 5 μm, 250 × 4.6 mm Silica (2) 100ALC Column (Phenomenex, Torrance, CA, USA) during a 10 min isocratic analysis run with a tetrahydrofurane—*n*-hexane mixture (4:96, *v*/*v*) as a mobile phase (flow rate: 1.3 mL/min). The column was thermostated at 35 °C. The injection volume was 5 μL. The fluorescence detector was set at 290 nm excitation wavelength and 330 nm emission wavelength. The calibration curve was made for each tocopherol standard (α-, β-, γ- and δ-tocopherol, treated as samples), and it was used for identification and quantification (Appendix A). Experiments were performed in triplicate. The results were expressed as mg of tocopherol per g of cherry seed oil (mg/g oil).

#### 2.4.3. DPPH Assay

Cherry seed oil scavenging capacity towards 1,1-diphenyl-2-picrylhydrazyl (DPPH) radicals was measured by the method described by Brand-Williams et al. [25] with slight modifications for lipid samples [26]. A freshly prepared methanol solution of DPPH (65 µM) was adjusted with methanol to reach 0.70 (±0.02) absorbance. Samples were previously diluted in ethyl acetate in a concentration of 50 mg/mL and mixed with DPPH reagent (0.1 mL and 2.9 mL, respectively) in glass tubes and incubated protected from light for 60 min. Freshly prepared Trolox aqueous solutions (0–0.8 mM, *R*^2^ = 0.987) were used to obtain the calibration curve (Appendix A). The concentration of DPPH reagent for the calibration curve was in range from 6.25 to 200 mg/L. The standard solution absorbance measurements were conducted in triplicates. Free radical scavenging measurements were performed at 517 nm in triplicates using UV-VIS spectrophotometer (6300 Spectrophotometer, Jenway, Cole-Parmer Ltd., St Neots, UK). The obtained results were expressed as µM of Trolox equivalents (TE) per g of cherry seed oil (µM TE/g).

#### 2.4.4. ABTS Assay

The capacity of cherry seed oil to scavenge ABTS free radicals was measured using the modified method [27]. ABTS stock solution was freshly prepared from mixture (1:1, *v*/*v*) of 2.45 mM potassium persulphate aqueous solution and 7 mM ABTS (2,2′-azino-bis-(-3-ethylbenzothiazoline-6-sulfonic acid) diammonium salt) aqueous solution and left in a dark place at room temperature for 16 h. ABTS reagent was made by dilution of the stock solution using 96% ethanol to reach 0.70 (±0.02) absorbance; 0.1 mL of sample (previously diluted in ethyl acetate to achieve 50 mg/mL concentration) and 2.9 mL of ABTS reagent were mixed and incubated for 5 h in a dark place. Freshly prepared Trolox aqueous solutions (0–0.8 mM, *R*^2^ = 0.987) were used to obtain the calibration curve (Appendix A). The concentration of the ABTS reagent for the calibration curve was in range from 6.25 to 200 mg/L. The standard solution absorbance measurements were conducted in triplicates. Absorbance was measured at 734 nm in triplicates using a UV-VIS spectrophotometer (6300 Spectrophotometer, Jenway, Cole-Parmer Ltd., St Neots, UK). The results were expressed as µM of Trolox equivalents (TE) per g of cherry seed oil (µM TE/g).

#### 2.4.5. Statistical Analysis

ANOVA and Tukey’s test were conducted to compare mean values and determine the differences between SFE parameters as well as differences between extraction techniques. Differences were considered significant if *p* < 0.05.

## 3. Results and Discussion

### 3.1. Extraction Yield

The supercritical fluid extraction yield of cherry seed oil was determined using the one-factor-at-time (OFAT) approach to define the impact of the extraction parameters. The yield of cherry seed oil ranged from 2.50% to 13.02% (Table 2). The highest extraction yield was obtained with SFE (350 bar, 70 °C, 0.4 kg/h; <800 µm), while oil yield with Soxhlet extraction and cold pressing were in the range from 3.19 to 5.15%. According to the literature reports, the yield of cherry seed kernels could be from 2.3% when oil is recovered from sour cherry seeds with an outer shell [28] or up to 17.0% when the outer shell was eliminated [29]. In our work, a pronounced increase in extraction yield was shown for particles smaller than 800 µm, since this fraction mainly consisted of the inner part of cherry seed that is richer in lipid molecules [6]. However, particularly small particles can be a reason for difficulties in supercritical CO_2_ extraction because of the complex penetration of SC-CO_2_ into the matrix and the decrease in extraction yield [30]. Since the particle had a pivotal role in extraction yield increase, it is highly possible that Soxhlet extraction would have higher oil yield if the experiment was performed with a sample fraction with a particle size <800 µm. Thus, the difference in extraction yield between SFE and Soxhlet extraction would be significantly smaller. From other SFE parameters, pressure and temperature had a positive effect on extraction yield causing its direct increase. Pressure increase enhances oil solubility, yet temperature increase weakens CO_2_ density. A cumulative impact of these parameters is possible after reaching cross-over pressure [31].

### 3.2. Fatty Acid Profile

The content of fatty acids in the cherry seed oil recovered by the SFE technique is shown in Table 2. A total of 11 fatty acids with a content higher than 0.02% were identified. Obtained results point to the presence of saturated, monounsaturated and polyunsaturated fatty acids in the cherry seed oil. The content of saturated fatty acids was between 10.06–10.36% of total fatty acids. Among the saturated fatty acids, palmitic (6.56–8.00%) and stearic (2.21–2.30%) were predominant. Additional saturated fatty acids are identified were myristic (0.05–0.07%), arachidic (0.72–1.10%) and behenic acid (0.23–0.31%). Monounsaturated fatty acids were present in higher percentages (41.96–42.72%), among which oleic acid was the most abundant with its content between 40.89 and 41.65%. Other monounsaturated fatty acids, such as palmitoleic and eicosenoic acid, were observed in significantly smaller amounts (<1%) in comparison with oleic acid. The content of polyunsaturated fatty acid was 47.22–47.68%. Linoleic acid (46.32–47.29%) prevailed compared to other identified acids. It could be noted that the applied extraction technique did not contribute to considerable differences between fatty acid profiles of oil samples recovered by cold pressing, Soxhlet extraction and SFE (Table 2). Comparable findings were reported by Górnaś et al. [33] after having investigated the fatty acid profile of sour cherry seed extracts recovered from different sour cherry cultivars using *n*-hexane. The most abundant fatty acids were oleic (25.25–45.30%) and linoleic (35.50–46.06%), followed by palmitic, α-eleostearic, stearic and arachidic acid. The other acids, present in less than 1%, were palmitoleic, α-linolenic and gondoic acid. Yılmaz and Gökmen [29] obtained sour cherry seed oil with *n*-hexane and supercritical CO_2,_ and the results showed that oleic acid was present in the highest amount (46.3%), followed by linoleic acid (41.5%). The fatty acids present in a lower amount were palmitic acid (6.4%), linolenic acid (4.6%) and stearic acid (1.2%). Linoleic and oleic acid are two of the main fatty acids in apple seed oil, grape seed oil, açaí berry oil and black cumin oil as well [22,34,35,36]. Oils rich in unsaturated fatty acids can have a beneficial impact on reducing levels of low-density lipoproteins (LDL) cholesterol and help in treatment of heart diseases [37]. Oils with a larger amount of polyunsaturated fatty acids (PUFA) are highly desirable for pharmaceutical and cosmetic applications [38]. Implementation of oleic acid into everyday nutrition can help in lessening the low-density cholesterol in plasma and decrease the possibility of cardiovascular disorders. Monounsaturated fatty acids are less likely to be susceptible to oxidative reactions, contrary to polyunsaturated fatty acids [39]. Linoleic acid belongs to essential PUFAs, and it has anti-carcinogenic properties and can prevent heart illnesses and autoimmune conditions, as well as skin diseases. Its hydrating characteristics make linoleic acid desirable for use in cosmetic products since it can moisturize and provide elasticity to the skin [40].

It is important to determine the functional quality of oil by calculating the PUFA/SFA ratio, as well as the hypocholesterolemic and hyperholesterolemic fatty acid ratio (h/H), the atherogenicity index (AI) and the thrombogenicity index (TI) [23]. The PUFA/SFA ratio of cherry seed oils was in the range between 4.60 and 4.69 (Table 3), which makes it acceptable for a human diet since oils with a PUFA/SFA ratio below 0.45 are indicated as unsuitable for a human diet because of the possible danger of raising cholesterol levels in blood [41]. Cherry seed oil (CSO) has a higher PUFA/SFA ratio than wheat germ oil [23]. It was noted that CSO h/H was from 6.17 to 6.61; therefore, CSO can be taken into account for use in a human diet. It is similar to olive oil but lower than linseed and sesame oil [42]. The highest level was noted in samples obtained by cold pressing, while the lowest level was in samples recovered by SFE. The atherogenicity index (AI) is useful to show linkage between fatty acids labeled as proatherogenic and antiatherogenic unsaturated fatty acids. Furthermore, the thrombogenicity index (TI) is important to demonstrate a predisposition to develop blood clots inside the vascular system [43]. In cherry seed oil samples, AI and TI were not significantly influenced by extraction technique and SFE extraction parameters. AI was in a range from 0.28 to 0.30, similar to pumpkin seed oil (0.34) and higher than sweet cherry seed oil (0.15) [44] and wheat germ oil (0.22–0.27) [23]. The TI index was 0.19–0.20, lower than in sesame, olive, bacaba and raspberry seed oil [42,45,46].

### 3.3. Tocopherols Content

Tocopherols comprise a chromanol ring core linked to a 16-carbon side tail. Depending on the number and placement of the methyl group on chromanol ring, tocopherols can exist as α-, β-, γ- and δ- isomers [47]. They represent amphiphilic compounds consisting of a lipophilic side chain linked to a lipid cell membrane, while the hydrophilic part is oriented towards the inner part of the cell. This may be the reason for the more complex extraction of tocopherols than oil and cause impoverishment of plant oils from important bioactive molecules [48].

The tocopherol content of individual and total tocopherols was read from the chromatograms made for each sample. Appendix A represents the chromatogram from the tocopherol analysis of the SFE performed at 350 bar, 70 °C and 0.3 kg CO_2_/h. The applied pressure of 200 bar (60.61 mg/100 g cherry seed oil) had the most positive effect on total tocopherol content when temperature, flow rate and sample particle size were held constant (Table 4). At 275 bar, total tocopherol content was lower (47.90 mg/100 g cherry seed oil), whereas it was the lowest at 350 bar (16.63 mg/100 g cherry seed oil). Regarding the individual tocopherol content, the γ-tocopherol yield was highest at 275 bar (30.36 mg/100 g cherry seed oil) but the lowest at 350 bar (14.27 mg/100 g cherry seed oil). A significantly lower amount of α- and δ-tocopherol was observed, and a pressure of 200 bar gave the highest yield of α-tocopherol, while 275 bar was the most favorable for δ-tocopherol (Table 4). The content of β-tocopherol was negligible compared to other tocopherols. The trend of tocopherol extractability can be explained by the higher vitamin E distribution on the seed surface; thus, it can be easily extracted in the first extraction phase, which can clarify its high content in low-yield extracts (Table 2). Extraction of the oil from undamaged cells is less efficient in terms of its tocopherol enrichment because of the complex intraparticular diffusion [49]. Supercritical CO_2_ at reduced pressure can lead the extraction process towards the selective extraction of tocopherols, but a further pressure increase can contribute to simultaneous extraction of other compounds present in the raw material and decrease tocopherol yield. This was confirmed in this study as well since total tocopherol content was highest at 200 bar (Table 4). At low pressure, oil can be rich in an unsaponifiable fraction contributing to better extraction of lipid-like compounds, including tocopherols, and in that way, tocopherol yield may be enhanced. In addition, another advantage of pressure reduction is a decrease of capital expenses of the SFE process [39,50].

The yield of individual and total tocopherols decreases proportionally with the temperature increase, and it was observed that the highest yields were achieved at 40 °C (42.56 mg/100 g cherry seed oil), yet the lowest were obtained at 70 °C (16.63 mg/100 g cherry seed oil) (Table 4). Temperature can play either a positive or a negative role in the SFE of tocopherol-rich oil since its increase causes the decrease of the solvent density and lower tocopherol yield when the applied pressure is constant [48]. Raising the temperature at a reduced pressure can increase CO_2_ solvating power and induce an increase of the vapor pressure of significant compounds which makes them preferable for extraction in the oil. However, supercritical CO_2_ density can be lower at an increased temperature and reduce SC-CO_2_ solubility [40,51,52]. It has been shown in a study by Benito-Román et al. [53] that tocopherols are not thermosensible, but high temperatures should be avoided to prevent losing the antioxidant activity of the extract.

The CO_2_ flow rate influenced the individual and total tocopherol content in a similar manner as temperature; therefore, the yield decreased with the flow rate increase. The CO_2_ flow rate of 0.2 kg/h contributed to the highest tocopherol content (Table 4). Concerning the particular tocopherols, γ-tocopherol content was remarkably higher, compared to α-, β- and δ-tocopherol content. Teslić et al. [23] observed a similar trend during the SFE of wheat germ oil, since CO_2_ flow rate increase negatively influenced tocopherol content, and this occurrence can be associated with the extraction of other related lipid molecules.

The influence of particle size on tocopherol content was not investigated in the case of the SFE of cherry seed oil. However, the literature reports the significance of properly prepared raw material before the extraction because it helps in mass transfer, but too small particles can cause channeling and aggravate CO_2_ penetration into the matrix [54]. In this study, it was possible to increase total carotenoid yield 3.5-fold from a fraction smaller than 800 µm compared to another fraction larger than 800 µm (Table 4). Pulverization has proven useful to achieve higher carotenoid extraction yield of paprika [55]. Terpenes and cannabidiol extraction from hemp flowers was also enhanced with a decrease in particle size, expanding the surface contact area, shortening diffusion distance and reducing internal mass transport [56].

Among the tocopherols in cherry seed oil obtained by different extraction techniques, γ-tocopherol was the predominant one, ranging from 12.37 to 40.60 mg/100 g oil. It was followed by α-tocopherol (1.75–10.54 mg/100 g oil) and δ-tocopherol (2.08–8.42 mg/100 g oil). The lowest amount was noted for β-tocopherol, and it was between 0.78 and 1.12 mg/g oil. These results are in accordance with the study by [57], though this work has shown the differences between eight sour cherry cultivars. It is noteworthy that cherry seed oil has higher total tocopherols content than red and white grape seed oil, chia seed oil and poppy seed oil [22,30,58]. For these reasons, cherry seed oil could be a potentially interesting source of vitamin E.

The best performing CO_2_ extract had a significantly higher amount of total tocopherols (60.61 mg/100 g oil), while cold pressing was comparable to Soxhlet extraction with methylene chloride. A similar observation was made by Benito-Román et al. [53] in the study which compared Soxhlet extraction with hexane and supercritical CO_2_ extraction of quinoa oil. The SFE extract had higher content of tocopherols compared to the hexane extract. Similarly, the increase in tocopherol content was noted in the work by Sun et al. [48], obtaining higher tocopherol yield in the SFE process compared to Soxhlet extraction with hexane. In the study from Ruttarattanamongkol et al. [39], cold pressing, Soxhlet extraction and SFE of *Moringa oleifera* seed oil were compared. No major difference between the tocopherol content was observed, but SFE on lower pressure provided higher tocopherol content than other extraction techniques.

### 3.4. Antioxidant Activity

The basis of the healing effect of natural products is their antioxidant activity. Due to the antioxidant potential, natural products have the ability to protect the cell from damage caused by free radicals and also represent the first barrier and foundation of defense against many degenerative conditions, such as cancer, heart disease, etc. [59]. In addition, the ability to protect cells and tissues from oxidative damage is the reason for their increasingly frequent use not only in the pharmaceutical industry but also in the food industry, where they are successfully used to protect food from oxidation and premature spoilage [60]. Components that contribute to the antioxidant power of natural products are mainly secondary metabolites of plants, such as polyphenols, but also components contained in essential oil as well as lipophilic compounds. All these components have a different mechanism of action as well as a different capacity, and the overall antioxidant activity of the product is affected by different factors. One of the factors that may influence the antioxidant activity of the extracts by causing the differences in their composition and content of antioxidant molecules are extraction techniques as well as extraction solvents [61]. In the frame of this paper, antioxidant activity of cherry seeds extracts was determined by two tests whose parallel application provides a deeper insight into the antioxidant capacity of the extracts. The obtained results are presented in Table 5.

In terms of DPPH test, the obtained results show that extracts made by Soxhlet (4.39 and 4.42 µM Trolox/g oil) and SFE (4.54 µM TE/g oil) extraction have similar ability to neutralize free DPPH radicals, while oil prepared by cold pressing has a slightly higher activity (5.21 µM TE/g oil). On the other hand, in the case of the ABTS assay, more considerable differences were observed in the activity of the tested samples. Namely, the cold-pressed oil (3.62 µM TE/g oil) showed a similar tendency towards the neutralization of ABTS radicals as the extract obtained by Soxhlet extraction using hexane (3.72 µM TE/g oil), and their activities were twice as low as those recorded in the case of the SFE extract (7.50 µM TE/g oil) and the extract obtained with methylene chloride (8.52 µM TE/g oil). Taking into account the results of both performed assays, it can be concluded that the SFE extraction technique offers advantages over other applied techniques and solvents. Although all observed extracts possess a satisfactory degree of anti-radical protection, the process of obtaining SFE extracts involves the use of a non-toxic and environmentally friendly solvent, which marks this extract as “green”. In addition, SFE extracts belong to the group of ready-to-use extracts, so their further application does not require additional purification or removal of solvent traces, and they can be directly incorporated into the desired product. In addition, the process of performing SFE extraction required less time than the other two processes, which makes it a time-effective process.

In comparison to the literature data [57], the obtained results are slightly lower, which can be explained by the differences in the cherry variety, the climatic region and the soil where the plant grew, as well as the solvent used during the extraction (diethyl ether). Nevertheless, taking into account the results of both conducted tests, as well as the advantages of using green extraction technologies, it can be concluded that cherry seed extracts obtained by the SFE technique represent a promising component in the formulation of various functional products. The inclusion of cherry seeds in the formulation of foods is also encouraged by the literature data on the toxicological effects of cherry seed extracts, which showed the absence of harmful effects on human health, keeping the kidney and liver functions intact [62]. Ishak et al. [30] compared commercial cold-pressed oil, SFE and Soxhlet extract in terms of DPPH assay and showed that CO_2_ extract and Soxhlet extract had better antioxidant potential towards DPPH radical than cold-pressed oil. This can be attributed to higher tocopherol content in these extracts. Pressure and temperature can correspondingly make an impact on antioxidant activity and tocopherol content, but it is possible that tocopherols, together with other antioxidant compounds could have a synergistic effect [63].

## 4. Conclusions

The emphasis of recent research has been on finding solutions that can cope with the constant rise of agro-industrial waste, minimizing or avoiding the use of dangerous chemical solvents and promoting the use of alternative options to provide high quality extracts. Cherry seed oil is rich in valuable biologically active compounds and can be utilized in the food, pharmaceutical and cosmetics industries. Supercritical fluid extraction (SFE) represents an excellent alternative to conventional extraction techniques; thus, it can lead to cherry seed oil’s further application in other industries. Optimization of the SFE process may be performed to selectively increase the yield of cherry seed oil. In this work, it was shown that it is possible to recover from 2.50 to 13.02% of oil from sour cherry seed, depending on the chosen extraction technique and process parameters. Supercritical extraction parameters, such as pressure, temperature and flow rate had an influence on tocopherol yield, and it can be concluded that extraction at low pressure, temperature and flow rate had a positive impact on tocopherol yield in oil. Additionally, particle size was an important parameter for SFE, since pulverization of the sample can lead to higher extraction yield and tocopherol yield as well. The SFE process could provide antioxidant-potent extracts without the need for purification. This can be taken into account in the case of optimization of the SFE process towards selectively recovering desired bioactives and obtaining high-value-added products. In comparison with other extraction techniques, it is possible to utilize the SFE process to recover “green” extracts free from harmful chemicals.

## Figures and Tables

**Table 1 foods-12-00011-t001:** Extraction techniques and process conditions.

Sample Name	Technique	Extraction Conditions			
CSO-CP	CP	26 Hz, t = 70 °C			
CSO-Hex	SE	Solvent: *n*-hexane, 1:4, 6 h			
CSO-MH		Solvent: methylene chloride, 1:4, 6 h			
	SFE	Factor 1 Pressure (bar)	Factor 2 Temperature (°C)	Factor 3 Flow rate (kg CO_2_/h)	Particle size (µm)
CSO-SFE-1		350	70	0.3	741
CSO-SFE-2		275	70	0.4	741
CSO-SFE-3		350	55	0.4	741
CSO-SFE-4		350	70	0.4	741
CSO-SFE-5		350	40	0.4	741
CSO-SFE-6		200	70	0.4	741
CSO-SFE-7		350	70	0.2	741
CSO-SFE-8		350	70	0.4	<800
CSO-SFE-9		350	70	0.4	>800

CP—cold pressing, SE—Soxhlet extraction, SFE—Supercritical fluid extraction.

**Table 2 foods-12-00011-t002:** Oil yield and fatty acid composition of cherry seed oils recovered by different extraction techniques.

Parameter	Extraction Technique													
	SFE											CP	SE	SE
	Extraction Conditions													
	70 °C, 0.4 kg/h			350 bar, 0.4 kg/h			350 bar, 70 °C			350 bar, 70 °C, 0.4 kg/h		26 Hz, t = 70 °C	*n*-Hexane 1:4, 6 h	Methylene Chloride 1:4, 6 h
	Pressure Influence			Temperature Influence			CO_2_ Flow Rate Influence			Particle Size Influence				
	200 bar	275 bar	350 bar	40 °C	55 °C	70 °C	0.2 kg/h	0.3 kg/h	0.4 kg/h	<800 µm	>800 µm			
Oil yield (%)	3.98 *	4.50 *	5.54 *	4.93 *	5.31 *	5.54 *	2.50 *	3.86 *	5.54 *	13.02 *	3.62 *	4.00	3.19	5.15
Fatty acids (g/100 g cherry seed oil)														
Myristic acid (C14:0)	0.07	0.06	0.05	0.05	0.05	0.05	0.05	0.05	0.05	0.05	0.05	0.04	0.04	0.05
Palmitic acid (C16:0)	8.00	7.07	6.67	6.56	6.59	6.67	6.92	6.74	6.67	6.91	6.71	6.35	6.55	6.62
Palmitoleic acid (C16:1)	0.44	0.40	0.38	0.38	0.39	0.38	0.39	0.39	0.38	0.37	0.38	0.38	0.38	0.37
Stearic acid (C18:0)	2.30	2.23	2.24	2.21	2.21	2.24	2.24	2.22	2.24	2.29	2.28	2.27	2.19	2.21
Oleic acid (C18:1n9c)	40.89	41.29	41.15	41.33	41.65	41.15	41.11	41.21	41.15	40.80	41.61	41.92	41.44	41.46
Linoleic acid (C18:2n6c)	46.32	46.83	47.29	47.26	47.26	47.29	47.16	47.25	47.29	47.37	46.68	46.82	47.17	47.00
Arachidic acid (C20:0)	0.94	1.03	1.10	1.10	0.72	1.10	1.04	1.08	1.10	1.09	1.14	1.10	1.10	1.12
Eicosenoic acid (C20:1n9)	0.43	0.43	0.42	0.43	0.43	0.42	0.43	0.41	0.42	0.43	0.42	0.43	0.43	0.43
α-Linolenic acid (C18:3n3)	0.33	0.33	0.32	0.32	0.33	0.32	0.33	0.33	0.32	0.32	0.33	0.33	0.32	0.33
Eicosadienoic acid (C20:2n6)	0.06	0.06	0.07	0.07	0.07	0.07	0.06	0.07	0.07	0.07	0.07	0.07	0.07	0.09
Behenic acid (C22:0)	0.23	0.28	0.31	0.30	0.30	0.31	0.28	0.25	0.31	0.30	0.32	0.30	0.32	0.32

*—Dimić et al. [32] CP—cold pressing SE—Soxhlet extraction.

**Table 3 foods-12-00011-t003:** Functional quality indices of cherry seed oils recovered by different extraction techniques.

Parameter	Extraction Technique													
	SFE											CP	SE	SE
	Extraction Conditions													
	70 °C, 0.4 kg/h			350 bar, 0.4 kg/h			350 bar, 70 °C			350 bar, 70 °C, 0.4 kg/h		26 Hz, t = 70 °C	*n*-hexane 1:4, 6 h	Methylene Chloride 1:4, 6 h
	Pressure Influence			Temperature Influence			CO_2_ Flow Rate Influence			Particle Size Influence				
	200 bar	275 bar	350 bar	40 °C	55 °C	70 °C	0.2 kg/h	0.3 kg/h	0.4 kg/h	<800 µm	>800 µm			
Functional quality indices														
SFA	11.53	10.66	10.36	10.22	9.87	10.36	10.52	10.34	10.36	10.50	10.65	10.06	10.19	10.31
MUFA	41.76	42.12	41.96	42.13	42.47	41.96	41.93	42.02	41.96	42.42	41.60	42.72	42.25	42.26
PUFA	46.70	47.22	47.68	47.65	47.66	47.68	47.55	47.64	47.68	47.08	47.76	47.22	47.56	47.42
UFA	88.47	89.34	89.64	89.78	90.13	89.64	89.48	89.66	89.64	89.50	89.35	89.94	89.81	89.69
Ratio S/U	0.13	0.12	0.12	0.11	0.11	0.12	0.12	0.12	0.12	0.12	0.12	0.11	0.11	0.11
AI	0.36	0.32	0.30	0.29	0.29	0.30	0.31	0.30	0.30	0.30	0.31	0.28	0.29	0.30
TI	0.23	0.21	0.20	0.19	0.19	0.20	0.20	0.20	0.20	0.20	0.20	0.19	0.19	0.19
h/H	5.11	5.84	6.17	6.31	6.32	6.17	5.95	6.11	6.17	6.21	5.90	6.61	6.34	6.27
PUFA/SFA	4.05	4.43	4.60	4.66	4.83	4.60	4.52	4.61	4.60	4.48	4.49	4.69	4.67	4.60

CP—cold pressing SE—Soxhlet extraction.

**Table 4 foods-12-00011-t004:** Tocopherol yield of cherry seed oils recovered by different extraction techniques.

Parameter	Extraction Technique													
	SFE											CP	SE	SE
	Extraction Conditions													
	70 °C, 0.4 kg/h			350 bar, 0.4 kg/h			350 bar, 70 °C			350 bar, 70 °C, 0.4 kg/h		26 Hz, t = 70 °C	*n*-Hexane 1:4, 6 h	Methylene Chloride 1:4, 6 h
	Pressure Influence			Temperature Influence			CO_2_ Flow Rate Influence			Particle Size Influence				
	200 bar	275 bar	350 bar	40 °C	55 °C	70 °C	0.2 kg/h	0.3 kg/h	0.4 kg/h	<800 µm	>800 µm			
Tocopherol (mg/100 g cherry seed oil)														
α-tocopherol	10.54 ± 1.97 ^f^	7.60 ± 0.45 ^e^	3.29 ± 0.31 ^a,b^	6.84 ± 0.33 ^e^	5.09 ± 0.84 ^b,c,d^	3.29 ± 0.31 ^a,b^	7.75 ± 0.10 ^e^	3.70 ± 0.53 ^a,b^	3.29 ± 0.31 ^a,b^	6.25 ± 0.34 ^c,d,e^	1.75 ± 0.17 ^a^	6.39 ± 0.08 ^c,d,e^	7.13 ± 0.18 ^e^	4.71 ± 0.28 ^b,c^
β-tocopherol	1.05 ± 0.03 ^b,c^	1.11 ± 0.11 ^c^	0.78 ± 0.07 ^b^	1.06 ± 0.13 ^b,c^	0.92 ± 0.07 ^b,c^	0.78 ± 0.07 ^b^	1.04 ± 0.01 ^b,c^	0.85 ± 0.08 ^b,c^	0.78 ± 0.07 ^b^	1.00 ± 0.06 ^b,c^	tr ^a^	1.07 ± 0.25 ^b,c^	1.12 ± 0.09 ^c^	0.93 ± 0.09 ^b,c^
γ-tocopherol	40.60 ± 7.74 ^e^	32.26 ± 1.79 ^d,e^	12.37 ± 1.50 ^a^	28.61 ± 1.38 ^c,d^	21.50 ± 4.21 ^c^	12.37 ± 1.50 ^a^	30.89 ± 0.60 ^d^	14.32 ± 2.42 ^a,b^	12.37 ± 1.50 ^a^	23.85 ± 1.46 ^c,d^	6.23 ± 1.28 ^a^	25.22 ± 0.05 ^c,d^	31.47 ± 0.86 ^d^	26.06 ± 1.56 ^c,d^
δ-tocopherol	8.42 ± 1.46 ^f^	6.93 ± 0.42 ^e,f^	3.01 ± 0.16 ^a^	6.05 ± 0.33 ^c,d,e^	4.72 ± 0.64 ^b,c^	3.01 ± 0.16 ^a^	6.53 ± 0.12 ^d,e^	3.35 ± 0.37 ^a,b^	3.01 ± 0.16 ^a^	5.13 ± 0.25 ^c,d^	2.08 ± 0.20 ^a^	5.41 ± 0.18 ^c,d^	6.64 ± 0.17 ^d,e^	5.68 ± 0.18 ^c,d,e^
Total tocopherols	60.61 ± 11.21 ^f^	47.90 ± 2.76 ^e^	19.45 ± 1.91 ^a,b^	42.56 ± 2.18 ^d,e^	32.22 ± 5.62 ^c,d^	19.45 ± 1.91 ^a,b^	46.22 ± 0.81 ^e^	22.22 ± 3.33 ^b,c^	19.45 ± 1.91 ^a,b^	36.22 ± 2.10 ^d,e^	10.06 ± 1.57 ^a^	38.09 ± 0.54 ^d,e^	46.36 ± 1.18 ^e^	37.37 ± 1.93 ^d,e^

The results are presented as mean ± SD, n = 3. Different letters within the same row indicate significant differences between applied extraction techniques according to Tukey’s HSD test (*p* < 0.05). CP—cold pressing SE—Soxhlet extraction.

**Table 5 foods-12-00011-t005:** Antioxidant activity of cherry seed oils recovered by different extraction techniques.

Parameter	Extraction Technique													
	SFE											CP	SE	SE
	Extraction Conditions													
	70 °C, 0.4 kg/h			350 bar, 0.4 kg/h			350 bar, 70 °C			350 bar, 70 °C, 0.4 kg/h		26 Hz, t = 70 °C	*n*-Hexane 1:4, 6 h	Methylene Chloride 1:4, 6 h
	Pressure Influence			Temperature Influence			CO_2_ Flow Rate Influence			Particle Size Influence				
	200 bar	275 bar	350 bar	40 °C	55 °C	70 °C	0.2 kg/h	0.3 kg/h	0.4 kg/h	<800 µm	>800 µm			
Antioxidant activity (µM TE/g oil)														
DPPH	3.40 ± 0.00 ^b,c^	6.22 ± 0.08 ^h^	4.54 ± 0.41 ^e^	2.18 ± 0.02 ^a^	3.74 ± 0.30 ^c,d^	4.54 ± 0.41 ^e^	5.57 ± 0.41 ^g,h^	4.70 ± 0.29 ^e,f^	4.54 ± 0.41 ^e^	2.99 ± 0.08 ^b^	4.92 ± 0.17 ^e,f,g^	5.21 ± 0.03 ^f,g^	4.39 ± 0.23 ^d,e^	4.42 ± 0.08 ^e^
ABTS	6.62 ± 0.19 ^b,c^	7.98 ± 0.05 ^e,f^	7.50 ± 0.14 ^d,e^	3.28 ± 0.19 ^a^	6.23 ± 0.32 ^b^	7.50 ± 0.14 ^d,e^	8.46 ± 0.22 ^f^	7.53 ± 0.10 ^d,e^	7.50 ± 0.14 ^d,e^	7.14 ± 0.20 ^c,d^	7.60 ± 0.27 ^d,e^	3.62 ± 0.10 ^a^	3.72 ± 0.15 ^a^	8.52 ± 0.09 ^f^

The results are presented as mean ± SD, n = 3. Different letters within the same column indicate significant differences between applied extraction techniques according to Tukey’s HSD test (*p* < 0.05). CP—cold pressing SE—Soxhlet extraction.

## Data Availability

Not applicable.

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
