# Peer review of "Isolation of Cherry Seed Oil Using Conventional Techniques and Supercritical Fluid Extraction"

_foods, 2022, doi:10.3390/foods12010011_

Round 1

Reviewer 1 Report

My main comments:

1 / please add information about the number of technological and apparatus repetitions

2 / Table 1 - illegible, please edit it - currently it is very chaotic

3 / If a curve was performed in the determination - please provide the equation, fit and concentration range of the analyte used for its preparation

4 / Please insert a chromatogram from the analysis of tocopherols: a) standards b) sample

5 / For all devices used, please write: name (model, manufacturer, country, city)

6 / Table 2 - no standard deviations, no statistics of homogeneous groups

7 / Tables 3 and 4 - edit please, it's impossible to read anything

Author Response

In the file.

Reviewer 2 Report

Isolation of cherry seed oil using convention a techniques and supercritical fluid extraction

Line 50-52: The oil content of sour cherry kernels….. needs a reference

Line 138: cold pressing: kindly state the quantity of cherry seed used for the extraction

Line 155: Cherry seeds (130.0±0.01 g g) delete one g

Line 167: kindly explain why yield was calculated using different quantity od cherry seed for each extraction method

Author Response

In the file.

Reviewer 3 Report

The idea of this paper is very interesting. The title of the manuscript is suitable of researching. Aim of the paper is clear. Introduction, Materials and Methods and Discussion are well written, but presentation of Results must be improved. The conclusion is connected with the obtained results and the contribution of the research is highlighted.

Check spelling and grammar throughout the paper. The tables are presented at a very low level of quality and need to be improved.

Author Response

In the file.

Round 2

Reviewer 1 Report

Accept in present form.